# Comparative Transcriptome Analysis Reveals Inhibitory Roles of Strigolactone in Axillary Bud Outgrowth in Ratoon Rice

**DOI:** 10.3390/plants13060899

**Published:** 2024-03-21

**Authors:** Wenzhen Ku, Yi Su, Xiaoyun Peng, Ruozhong Wang, Haiou Li, Langtao Xiao

**Affiliations:** 1Hunan Provincial Key Laboratory of Phytohormones and Growth Development, Hunan Agricultural University, Changsha 410128, China; wenzhenku@163.com (W.K.); yisu@hunau.edu.cn (Y.S.); bsfbhd@163.com (X.P.); wangruoz@163.com (R.W.); 2Hunan Provincial Key Lab of Dark Tea and Jin-Hua, College of Materials and Chemical Engineering, Hunan City University, Yiyang 413000, China

**Keywords:** ratoon rice, strigolactone, axillary bud outgrowth, phytohormone homeostasis, transcriptome analysis

## Abstract

Axillary bud outgrowth, a key factor in ratoon rice yield formation, is regulated by several phytohormone signals. The regulatory mechanism of key genes underlying ratoon buds in response to phytohormones in ratoon rice has been less reported. In this study, GR24 (a strigolactone analogue) was used to analyze the ratooning characteristics in rice cultivar Huanghuazhan (HHZ). Results show that the elongation of the axillary buds in the first seasonal rice was significantly inhibited and the ratoon rate was reduced at most by up to 40% with GR24 treatment. Compared with the control, a significant reduction in the content of auxin and cytokinin in the second bud from the upper spike could be detected after GR24 treatment, especially 3 days after treatment. Transcriptome analysis suggested that there were at least 742 and 2877 differentially expressed genes (DEGs) within 6 h of GR24 treatment and 12 h of GR24 treatment, respectively. Further bioinformatics analysis revealed that GR24 treatment had a significant effect on the homeostasis and signal transduction of cytokinin and auxin. It is noteworthy that the gene expression levels of *OsCKX1*, *OsCKX2*, *OsGH3.6*, and *OsGH3.8*, which are involved in cytokinin or auxin metabolism, were enhanced by the 12 h GR24 treatment. Taken overall, this study showed the gene regulatory network of auxin and cytokinin homeostasis to be regulated by strigolactone in the axillary bud outgrowth of ratoon rice, which highlights the importance of these biological pathways in the regulation of axillary bud outgrowth in ratoon rice and would provide theoretical support for the molecular breeding of ratoon rice.

## 1. Introduction

Rice is one of the main staple foods for human consumption all over the world. Since the new century, global agriculture has faced a series of challenges, such as global warming, reduced arable land, and population explosion. To solve the crisis of food shortages, it is crucial to improve crop productivity, especially for rice. For a higher net energy ration and benefit-to-cost ration, ratoon rice is considered to be an alternative system for rice cultivation [1]. Ratoon rice refers to a second rice crop that sprouts from stem nodes on the rice stubble left behind after the harvest of the main crop [2]. Ratoon rice grows 65% earlier than main crops and requires 50% to 60% less labor. The production cost is also lower than that of main crops due to the minimized cost for land preparation, transplantation, and crop maintenance. Ratoon rice requires less time, and its yield is up to 50% of the main crop’s, which increase the opportunity for cropping intensity per unit of cultivated area [2].

The production rate of ratoon rice depends on different factors such as variety selection, sowing time, stubble height, and the growth status of the ratoon spikelet [3]. The ratoon spikelet consists of an outgrowth of the axillary bud and tillering at the base. The outgrowth of the axillary bud could be regulated by genes related to the initiation of the axillary meristem, including *BLIND*, *LATERAL SUPPRESSOR*, *MONOCULM1* (*LAS*), and *REGULATOR OF AXILLARY MERISTEM* (*RAX*) genes in rice, tomato, and *Arabidopsis*, respectively [4,5]. Many transcription factors in both monocots and dicots, for example *TEOSINTE BRANCHED1* (*TB1*)/*FINE CULM1* (*FC1*)/*BRANCHED1* (*BRC1*)/*TEOSINTEBRANCHED1*/*CYCLOIDEA*/*PCF* (*TCP*), were reported as the key factor to promote cell proliferation arrest and control axillary bud outgrowth [6,7]. These transcription factors also perform critical roles in auxin (IAA) and strigolactone (SL) signaling pathways.

In addition, axillary bud outgrowth is also regulated by phytohormone signaling and crosstalk among phytohormones, such as IAA, cytokines (CTK), and SLs [8]. Previous studies have demonstrated that both CTK and SL function as second messengers of auxin in the regulation of axillary bud outgrowth, with an antagonistic relationship [9]. In particular, as a class of carotenoid-derived phytohormones, they would inhibit axillary bud growth [10]. SLs are synthesized with carotenoid as a precursor in plastids involving several enzymes, such as a carotene isomerase *DWARF 27* (*D27*) and two carotenoid cleavage dioxygenases, CAROTENOID CLEAVAGE DIOXYGENASE 7 (*CCD7*) and CAROTENOID CLEAVAGE DIOXYGENASE 8 (*CCD 8*), encoded by *MORE AXILLARY GROWTH 3* (*MAX3*)/*RAMOSUS 5* (*RMS5*)/*DECREASED APICAL DOMINANCE 3* (*DAD3*)/DWARF 17 (*D17*) and *MORE AXILLARY GROWTH 4* (*MAX4*)/*RAMOSUS 1* (*RMS1*)/*DECREASED APICAL DOMINANCE 1* (*DAD1*)/*DWARF 10* (*D10*), respectively [11]. The SL receptor DWARF 14 (D14), one member of the *α*/*β* hydrolase proteins, not only binds GR24 (analogue of SL) but also cleaves GR24 to produce the D-ring-derived Covalently Linked Intermediate Molecule (CLIM) that serves as the active form of SL and is covalently bound to the receptor [12,13]. Mutations in the genes of SL biosynthesis lead to dramatically increased tiller numbers in rice, and this phenotype can be suppressed by exogenous treatment with the commonly used SL analog GR24 [14,15,16,17]. However, the regulatory mechanism of SLs regulating axillary bud outgrowth in ratoon rice are less well reported.

High-throughput sequencing technology has played an important role in revealing the molecular mechanisms of various biological processes in organisms. RNA sequencing (RNA-Seq) technology has been widely applied to assist in determining differentially expressed genes (DEGs) involved in different biological processes in many species and may be a promising method to address the genes associated with axillary bud outgrowth in ratoon rice. However, no studies on axillary buds in ratoon rice using RNA-Seq technology have been documented so far, and the molecular mechanism regulating axillary bud outgrowth in ratoon rice is poorly understood. In the present study, to better reveal whether SLs are involved in regulating axillary bud outgrowth in ratoon rice, we used RNA-seq to identify genes in ratoon rice associated with axillary bud outgrowth after the application of GR24, which is widely used in the study of the mechanism of SLs controlling shoot branching in various plants, such as rice [18], pea [7], tomato [19], and tobacco [20]. The effect of GR24 treatment on the outgrowth of axillary buds including DEGs, phenotypes, and contents of other phytohormones was also comprehensively analyzed. This approach may provide data support for studying the axillary bud growth network of ratoon rice jointly regulated by SL, IAA, and CTK.

## 2. Results

### 2.1. Strigolactone Inhibits the Outgrowth of Axillary Buds after the Harvest of Seasonal Rice

In this study, we investigated the dynamic changes in outgrowth of axillary buds through spraying a GR24 treatment on the plants after the harvest of the first seasonal rice (Figure 1A,B). Our results showed that the second buds from the upper spike in the control group began to germinate quickly after the harvest of the first seasonal rice, which significantly promoted the length of the second buds (Figure 1B,C). Meanwhile, the inhibitory effect of GR24 treatment on the second buds’ outgrowth could be observed within 1 d, and the length of the second buds was significantly lower than that of the control group at the same period, and the difference disappeared after seven days (Figure 1C). The ratoon rates after GR24 treatments were consistently lower than the control group within 30 days (Figure 1D). And the most significant inhibitory effect was observed at 3 and 7 days after GR24 treatment. The ratoon rates at 3 d and 7 d after GR24 treatments were only 0.19 and 0.33, respectively, while in the control they were 0.38 and 0.55, respectively (Figure 1D). These results indicated that GR24 could inhibit the outgrowth of ratoon rice via regulating ratoon rates, but not the length of the bud.IAA and CTK play important roles in the sprouting and outgrowth of axillary buds. We first investigated the pattern of phytohormone contents along with different organs and developmental stages. Results show that IAA accumulated particularly at the panicle across the booting stage, heading stage, graining stage, and maturity stage (Appendix A), while ZR accumulated in the node for both the milk-ripe stage and the maturity stage (Appendix A). Results of phytohormone contents showed that both IAA and Z were inhibited by at most 40% at 3 d by GR24 treatments in the second bud from the top of the first seasonal rice (Figure 2A–C), and ZR was inhibited by 75% under GR24 treatment (Figure 2B). IAA was inhibited by 76% at 3 d by GR24 treatment at the second section from the top of the first seasonal rice (Figure 2D), and ZR was inhibited by at most 50% at 3 d by GR24 treatment (Figure 2E). Consistently, the amount of Z was inhibited by at most 30%, induced by GR24 (Figure 2F). Combined with phenotypic changes in axillary buds under GR24 treatments, these results suggested that SL can regulate the axillary bud outgrowth of ratoon rice by affecting the accumulation of IAA and CTK.

### 2.2. Transcriptome Analysis on Differentially Expressed Genes Induced by Strigolactone

To further uncover the key genes that regulate the outgrowth of the axillary bud induced by GR24 in HHZ, we performed a transcriptome analysis. Results suggest that 27,184,878 raw reads and 26,635,485 on average in 15 samples were identified, with 94% Q30 value (Appendix A). The uniquely mapped reads accounted for 44.6% of the total mapped reads in the 15 samples (Appendix A). The samples of the control (referred to as CK) and GR24 treatment (referred to as GR24) at 0 h, 6 h, and 12 h were clustered together within each group (Figure 3A). We further analyzed the differentially expressed genes (DEGs) in different combinations of CK- and GR24-treated groups for different durations. In total, 1135 significantly differentially expressed genes (433 upregulated and 702 downregulated) were found in CK-6h vs. CK-0h. There were 285 upregulated and 613 downregulated DEGs in the CK-12h vs. CK-0h, 245 upregulated and 497 downregulated DEGs in the GR24-6h vs. CK-0h, and 1444 upregulated and 1433 downregulated DEGs in the GR24-12h vs. CK-0h (Figure 3B). Notably, the number of upregulated DEGs was less than the downregulated DEGs in the early period, and the maximum number of DEGs was in the GR24-12h vs. CK-0h group. In order to validate the RNA-seq data, nine DEGs were selected to confirm the expression through qPCR analysis (Appendix A). For these nine DEGs, the expression patterns in qPCR results were similar to RNA-Seq data, and were shown as a fragment per kilobase of transcript per million mapped reads (FPKM) values. The results also validate the reliability of RNA-seq data.

Based on the overlapping DEGs identified above by GR24-6h vs. CK-6h and GR24-12h vs. CK-12h, we performed GO and KEGG analysis. Results show that the single-organism process and nitrogen compound metabolic process, the intrinsic component of the membrane, the intracellular membrane-bounded organelle, nucleic acid binding transcription factor activity, and catalytic activity were significantly enriched in the list of DEGs for the two comparisons (Figure 4A). KEGG analysis suggested that metabolic pathways, the biosynthesis of secondary metabolites, carbon metabolism, protein processing in the endoplasmic reticulum, glycolysis/gluconeogenesis, carbon fixation in photosynthetic organisms, and phytohormone signal transduction occurred in the comparisons of both CK-12h vs. CK-0h and CK-6h vs. CK-0h (Figure 4B), which was also consistent for the comparisons of both GR24-6h vs. CK-0h and GR24-12h vs. CK-0h for both GO and KEGG analyses (Figure 4C). Further analysis revealed that the expression of DEGs in the KEGG enrichment pathways was mainly downregulated.

### 2.3. Comprehensive Analysis on the Expression of Phytohormone-Related Genes, Especially Those Involved in IAA and CTK Homeostasis

Since there are many biological pathways that were enriched in the DEGs list induced by GR24 based on GO and KEGG, we analyzed these DEGs using separated pathways, including SL, CTK, and IAA signaling pathways. Firstly, we determined the relative expression levels of some genes related to the SL signaling pathway by qPCR. As shown in Figure 5A, the expression levels of *D53*, which is an important repressor of SL signaling, were dramatically inhibited by GR24 (Figure 5A). Meanwhile, we found that *Ideal Plant Architecture 1* (*IPA1*), a key regulator of the plant architecture in rice, functions as a direct downstream component of D53 in regulating the tiller number and SL-induced gene expression, the expression of which is significantly induced within 12 h after GR24 treatment (Figure 5B). Furthermore, we found that the expression level of *FC1*, which works downstream of strigolactones to inhibit the outgrowth of axillary buds in rice, was also significantly higher than that of the control after GR24 treatments for 6 h and 12 h (Figure 5C). In addition to this, the relative expression of *D27* was also significantly induced after GR24 treatment (Figure 5D). Taken together, these results indicated that the treatment of GR24 was very effective.

It is known that IAA and CTK play important roles in regulating the outgrowth of the axillary bud in rice. Considering that the results of this study showed that GR24 treatment impacts the contents of IAA and CTK during axillary bud outgrowth, we comprehensively analyzed the expression of genes related to IAA and CTK homeostasis, including biosynthesis, transport, and metabolism, during axillary bud outgrowth in seasonal rice after GR24 treatment. The results indicated that most of the genes related to CTK and IAA homeostasis showed similar expression patterns with or without GR24 treatment. Therefore, the question is which differentially expressed genes are more worthy of attention? Firstly, the relative expression of *OsYUCCA5* (BGIOSGA037469), involved in IAA biosynthesis, was obviously repressed after GR24 treatment (Figure 6A). We also found that the relative expressions of *OsGH3.8* (BGIOSGA023979) and *OsGH3.6* (BGIOSGA018825) were significantly increased at 12 h after GR24 treatment (Figure 6A). These results suggest that SL could regulate IAA homeostasis via the repression of biosynthesis and the promotion of metabolism during the axillary bud outgrowth of ratoon rice. Additionally, the relative expression of *CKXs*, such as *OsCKX1* (BGIOSGA002233), *OsCKX2* (BGIOSGA002195), *OsCKX10* (BGIOSGA020998), important in CTK metabolism, were significantly increased at 12 h after GR24 treatment (Figure 6B), which suggested that SL could inhibit CTK accumulation via the stimulation of CTK metabolism during the axillary bud outgrowth of ratoon rice.

Transcription factors are indispensable in the promotion of cell proliferation arrest and the control of axillary bud outgrowth. There were 62 transcription factors that significantly altered in different comparisons of CK and GR24 groups for different durations (Appendix A). In particular, there were 17 DEGs related to the AP2/ERF gene family, followed by WRKY, which had 6 DEGs (Appendix A). Interestingly, in the AP2/ERF gene family, many genes, such as *OsERF102* (BGIOSGA030907) and *OsERF20* (BGIOSGA008802), showed the most significant increase in gene abundance based on transcriptome analysis (Appendix A). These transcriptome data will provide powerful data for further revealing the gene network of SL-regulated axillary bud outgrowth in ratoon rice.

Collectively, our results provide new mechanistic insights into how SL regulates the axillary bud outgrowth of ratoon rice. On the one hand, SL can promote the expression of *OsFC1*/*OsTB1*, which can inhibit the outgrowth of the axillary bud directly. On the other hand, SL can indirectly suppress the accumulation of IAA and CTK via the repression of the expression of the IAA biosynthesis gene *YUCCA5* and the acceleration of the expression of the IAA metabolism gene *GH3* and CTK metabolism genes *CKXs*, and SL is then able to achieve the inhibition of axillary bud outgrowth in ratoon rice (Figure 7).

## 3. Discussion

Phytohormones, as signal substances in plants, play an important role in the sprouting and growth of buds. Previous studies have shown that phytohormones ABA (abscisic acid), CTK (cytokinins), and SL (strigolactone) are involved in the regulation of axillary bud growth [21,22,23]. However, far fewer studies have been reported about SL effects on the outgrowth of axillary buds in ratoon rice. GR24, a synthetic SL analog, has been widely used to study multiple pathways by which SL affects various aspects of plant growth and development [18,19,20,24]. In this study, we observed that after treatment with GR24, SL significantly inhibits the ratoon rate (Figure 1A–D), indicating that SL is involved in the growth regulation of the middle and upper axillary buds of ratoon rice, and that its mechanism is similar to that of tiller regulation. Our study extends the knowledge of SL effects on the outgrowth of axillary buds in ratoon rice.

SLs are a group of terpenoid lactones which first became known for their function in rhizosphere parasitic and symbiotic interactions [25,26]. SLs have been identified as plant hormones that inhibit bud outgrowth in different plant species [10,14,15,16]. Deficiencies in SL biosynthesis and perception lead to the excessive outgrowth of axillary buds. Rice DWARF 53 (D53) is an important component involved in SL signaling, in which D53 acts as a substrate of the SCF^D3^ ubiquitination complex and functions as a repressor of SL signaling [18]. Recent research has reported that IPA1 was identified as a direct downstream component of D53 in regulating the tiller number and SL-induced gene expression [27]. Moreover, *D53* can interact with *OsBZR1* (*BRASSINZOLE RESISTANT 1*, *BZR1*) to inhibit the expression of tillering inhibitor *FC1*, thereby promoting rice tillering [28]. Overall, D53 promotes the outgrowth of axillary bud while IPA1 and FC1 take the opposite effect. Consistent with these reports, our study showed that the expression of *D53* was significantly downregulated and that of *IPA1* and *FC1* were greatly upregulated after GR24 treatment (Figure 5A), demonstrating that the SL signaling pathway also plays an important role in the outgrowth of axillary buds in ratoon rice.

IAA (indole-3-acetic acid) is synthesized in the apical meristem and transported to the basal tissue through polar transport to inhibit lateral bud growth. Previous studies indicated that the growth of axillary buds of gramineous crops was also affected by apical dominance by IAA [18,29]. Auxin also has inhibitory effects on bud development, through coordination with SL signals [30]. Auxin and SL signaling could interact in chain feedback loops. GR24 application or SL synthesis significantly reduce the basipetal transport of auxin [30,31,32]. In our study, the IAA content declined both in the second nodes and the section after GR24 treatments in first seasonal rice (Figure 2A,D), which may be achieved by the downregulation of biosynthesis and the upregulation of the metabolism (Figure 6A). Here, we demonstrated that the expression of *OsYUCC5* was significantly downregulated within 6 h after GR24 treatment (Figure 6A). Consistent with the finding of decreasing IAA levels in the outgrowth of axillary buds (Figure 2), it is likely that SL may constitutively regulate IAA content by affecting IAA homeostasis. These results were consistent with the auxin canalization model on bud growth regulation. More auxin biosynthesis and enhanced auxin transport in buds imply the formation of a new auxin source and the onset of efflux auxin to the stem for further outgrowth. The reduced expression of *OsYUCC5* confirmed local auxin biosynthesis in buds, which is necessary as an auxin source. The GRETCHEN HAGEN 3 (GH3) acyl acid amido synthetase family, pivotal in conjugating IAA with amino acids, has been proposed as part of an important auxin attenuation mechanism, based on its auxin-conjugating enzymatic activity and gain-of-function phenotypes [33,34]. Here, we found that the relative expressions of *OsGH3.8* and *OsGH3.6* in the axillary bud outgrowth were significantly induced after GR24 treatment (Figure 6A), which implies that SL may affect the IAA content in ratoon rice by promoting IAA metabolism. Meanwhile no obvious change was detected in the expression of IAA transporter genes, which suggests that the feedback loop between SL and IAA in the outgrowth of axillary buds in ratoon rice may be unique.

Previous studies have confirmed that SL and CTK have been shown to function antagonistically in pea bud elongation [7,35] and rice mesocotyl elongation in darkness [36]. Moreover, numerous studies have reported the crucial role of CTK in regulating tillering, mainly through affecting the CTK metabolism or signaling pathways. SL could induce the expression of *OsCKX9* to reduce CTK content and control the tiller number in rice [17]; a nucleoredoxin-encoding gene *RRA3*, which inhibits cytokinin signaling in ratooning bud growth, negatively regulates rice ratooning ability [37]. In this study, we found that SL can also suppress the accumulation of CTK via an acceleration in the expression of CTK metabolism genes, such as *OsCKX1*, *OsCKX2*, and *OsCKX10*, to inhibit axillary bud outgrowth in ratoon rice (Figure 6B). CKX enzymes are responsible for the catalysis of cytokinin degradation reaction in plants, and their activity ultimately leads to altered CTK contents in buds. Hence, it may be that SLs mediate CKX activity to regulate the CTK levels in buds in an ordinary way. Consistent with this, there were significantly decreased contents of CTK and IAA in the axillary bud outgrowth of ratoon rice after GR24 treatment (Figure 2). It should be mentioned that the interactions of SL with IAA and SL with CTK have been established and proven to be critical for plant development; for example, SL can inhibit auxin transport capacity and auxin can inhibit CTK biosynthesis. Therefore, our study reveals the interaction among strigolactones, auxin, and cytokinin in controlling the axillary bud outgrowth of ratoon rice.

Transcription factors (TFs) play an extremely important role in receiving various phytohormone signals and co-regulating plant growth and development [38,39,40]. Axillary bud outgrowth is an important developmental process regulated by various transcription factors. It has been reported that the WRKY transcription factor WRKY71/EXB1 can control shoot branching by transcriptionally regulating *RAX* genes in *Arabidopsis* [41]. Extensive studies have revealed that AP2/ERF family genes are involved in the formation and development of lateral roots and bud branching [42,43]. AP2/ERF transcription factor CmERF053 in chrysanthemum has a positive regulatory effect on the stem, lateral roots, and drought resistance, and it may be involved in the cytokinin-related bud branching pathway [44]. In this study, we identified many transcription factor families, such as the AP2/ERF gene family, WRKY, and bZIP. Most notably, 17 AP2/ERF family members were screened for differential expression, and we found that the two genes, *OsERF102* (BGIOSGA030907) and *OsERF20* (BGIOSGA008802), were significantly upregulated at 6 h and 12 h after GR24 treatment and were significantly higher than those in the control group. Among them, *OsERF102* was reported to regulate rice’s internode elongation [45]. All these supporting results suggested that transcription factors may play an important regulatory role in the axillary bud outgrowth of ratoon rice.

## 4. Materials and Methods

### 4.1. Materials and Growth Conditions

A conventional *indica* rice, Huanghuazhan (HHZ), was used in this study, and was donated by the Rice Germplasm Resource Bank of the Rice Research Institute of the Hunan Academy of Agricultural Sciences. HHZ is characterized by strong ratooning ability, high yield in production, fertilizer tolerance, and lodging resistance, and it is widely cultivated as a ratoon rice in southern China. Seeds were soaked for 48 h for germination, and sown in pots filled with moist soil in a greenhouse. The 1-month-old seedlings were transplanted to pots (30 cm × 23 cm × 22 cm), with two seedlings per plant and three plants for each pot, and 6 g of balanced base fertilizer (N:P:K = 15:10:10) was applied for each pot at 7 d before transplanting; 3 g of tillering fertilizer (N:P:K = 15:10:10) was applied to each pot at the tillering stage. Intercultural operations, i.e., weeding, water management, and plant protection measures, were followed when needed to maintain a normal growth of experimental rice plants.

### 4.2. GR24 Treatment on Axillary Buds

To analyze the phytohormone effects on the global gene expression of axillary buds, we sprayed 10 µM GR24 immediately after harvesting HHZ at the maturity stage (with a pile height of 40 cm) and the same amounts of sterile water were used as the control. Buds were sampled after 0 h, 6 h, and 12 h. We first peeled off the leaves and then the leaf sheaths, and finally used a blade to cut off the axillary buds located on the second node (second bud), as well as the second node. The second bud is the axillary bud at the main node where the axillary buds germinate into spikes during the rice regeneration process. For each treatment, 30 buds and nodes were collected, and then frozen in liquid nitrogen for 1 h, and then stored in a −80 °C refrigerator.

In addition, to examine the effects of GR24 on the growth dynamics of axillary buds, samples were also collected at 0 d, 1 d, 3 d, 7 d, 15 d, 22 d, and 30 d after GR24 treatment. Both the second node and second bud were cut off with a blade, as mentioned above, and the length of the bud was measured. For each treatment, 30 stalks in total were collected to measure the bud length, and a digital camera was used to record the outgrowth characteristics of axillary buds of the first seasonal rice. In this study, the regeneration rate was defined as the ratio calculated by the ratio of the plant growth of the ratoon against that of the previous crop, as reported previously [46].

### 4.3. Phytohormone Determinations

The extraction and determination of phytohormones, including indole-3-acetic acid (IAA), trans-Zeatin-riboside (ZR), and Zeatin (Z) content, were performed as previously documented [47]. Each sample included three biological replicates, and the determination was performed by an LC-MS/MS tandem mass spectrometer (8030Plus, Shimadzu, Kyoto, Japan). The specific steps are as follows:

Two hundred milligrams of each fresh sample was frozen with liquid nitrogen and homogenized well using a TissueLyser homogenizer (Qiagen, Shanghai, China). Following the addition of 1 mL 80% methanol, homogenates were mixed well in an ultrasonic bath (KQ3200E, Kunshan, China) and kept overnight at 4 °C. After centrifugation at 15,200× *g* for 10 min, the supernatant was collected and vacuumed to dryness. After the sample had been dissolved in 100 µL 10% methanol and centrifuged at 15,200× *g* for 15 min, 5 µL solution was injected into the LC-MS/MS system.

Liquid chromatography was performed using a 2 mm i.d. × 75 mm Shim-pack XR ODS I column (2.2 µm; Shimadzu) under a column temperature of 40 °C. The mobile phase, comprising solvent A (0.02% [*v*/*v*] aqueous acetic acid) and solvent B (100% [*v*/*v*] methanol), was employed in gradient mode, simply described as “min/%/%”, i.e., retention time/concentration of solvent A/concentration of solvent B. The gradients for IAA were 0/90/10, 6/40/50, 6.1/10/90, and 7/90/10, while the gradients for Z and ZR were 0/90/10, 6/40/50, 5.1/10/90, 5/10/90, and 6.1/80/20, respectively.

The mass system was set to multiple reaction monitoring mode using electrospray ionization for different hormones. For IAA, the negative ion mode was used. For Z and ZR, the positive ion mode was used. Operation conditions, including nebulizing gas flow, drying gas flow, desolvation temperature, and heat block temperature, were optimized for different hormones using standards. A collision energy of 16 eV and mass-to-charge ratio (*m*/*z*) of 147.2/130 were employed for IAA; a collision energy of 19 eV and mass-to-charge ratio (*m*/*z*) of 352.2/220.1 for ZR; and a collision energy of 25 eV and mass-to-charge ratio (*m*/*z*) of 225/137.1 for Z.

### 4.4. mRNA Extraction and Library Preparation

Total RNA was extracted using TRIzol reagent according to the manufacturer’s instructions (Invitrogen, Carlsbad, CA, USA). RNA degradation and contamination were monitored on 1% agarose gels, and purity was checked using the Nano-Photometer spectrophotometer (IMPLEN, Hercules, CA, USA). RNA integrity was assessed using the RNA Nano 6000 Assay Kit of the Agilent Bioanalyzer 2100 system (Agilent Technologies Inc., Carlsbad, CA, USA). A total amount of 1.5 μg RNA per sample was used as the input material for the RNA sample preparations. Sequencing libraries were generated using NEB Next Ultra RNA Library Prep Kit for Illumina (NEB, Lincoln, NE, USA), following manufacturer’s recommendations, and index codes were added to attribute sequences to each sample. The clustering of the index-coded samples was performed on a Bot Cluster Generation System using a HiSeq 4000 PE Cluster Kit (Illumina, Hercules, CA, USA), as reported previously [48]. After cluster generation, the library preparations were sequenced on an Illumina Hiseq 4000 platform and 150-bp paired-end reads were generated.

### 4.5. Read Mapping and Differentially Expressed Analysis

The quality of RNA-seq data (fastq files) was assessed by FastQC software (v 0.12.0) (http://www.bioinformatics.babraham.ac.uk/projects/fastqc/; accessed on 18 July 2023). The adaption and reads with low quality from raw RNA-seq reads were trimmed using trim_galore software (http://www.bioinformatics.babraham.ac.uk/projects/trim_galore/; accessed on 18 July 2023). The quantification of genes and isoforms was performed using cufflinks version 2.2.1. RNA-seq analysis was performed with STAR [49] software (version 2.5.3a; http://github.com/alexdobin/STAR; accessed on 15 March 2023) with the rice reference genome IRGSP-1.0, as well as a gene transfer format (GTF) file (downloaded from Ensembl Plants http://plants.ensembl.org/; accessed on 24 January 2022).

To identify differentially expressed genes (DEGs) between GR24 treatment and control at different hours after GR24 treatments, a fragment per kilobase of transcript per million mapped reads (FPKM) method was applied to calculate the transcript abundance. Notably, DEGs were determined by the R package ‘DESeq2’ [36] with the read counts reported by STAR [50]. Only genes with the adjusted *p*-value < 0.05 were considered as DEGs. To reduce transcription noise, each isoform/gene was included for analysis only if its FPKM values were >0.01, a value was chosen based on gene coverage saturation analysis, as described earlier [51].

### 4.6. GO and KEGG Analysis

For Gene Ontology (GO) annotation, we used an in-house Perl script UniProtKB GOA file (ftp.ebi.ac.uk/pub/databases/GO/goa; accessed on 15 July 2022). A KOBAS (KEGG Orthology Based Annotation System, v2.0) was applied to identify the reprogrammed biochemical pathways of each pathway, as previously documented [52]. Both GO and KEGG terms with a corrected *p*-value less than 0.05 were considered significantly enriched among the DEGs.

### 4.7. Quantitative Transcript Measurements

Based on transcriptomes analysis, qPCR was used to confirm the expressed pattern of genes involved in different phytohormone signals in response to GR24 in ratoon rice. The RNA sample used was the same as the transcriptome determinations. RNA extraction and reversed cDNA were performed as described previously [53]. The qPCR analysis was performed using SYBR Green PCR Master Mix (Applied Biosystems, Forster City, CA, USA) and a real-time PCR system (ABI StepOnePlus, Applied Biosystems, USA). Primers for qPCR were designed using Primer Prime Plus 5 Software Version 3.0 (Applied Biosystems, USA). Primers were listed in Appendix A. The qPCR running program consists of a reverse transcription step at 48 °C for 30 min and a Taq polymerase activation step at 95 °C for 30 s, followed by PCR: 45 cycles at 95 °C for 15 s, 61 °C for 20 s, and 72 °C for 30 s, followed by a melting cycle. Assays were performed with three biological samples from each treatment, and measurements were replicated three times. The *Actin1* gene was used as an expression control (housekeeping gene). The relative expression of a gene against *Actin1* was calculated as 2^−ΔΔCT^ (ΔCT = CT, gene of interest^−CT^), as described earlier [54].

## 5. Conclusions

In this study, GR24, as a synthetic SL analog, was successfully applied to mimic the effects of SL on axillary bud outgrowth in ratoon rice. Results show that the elongation of the axillary buds in first seasonal rice was significantly inhibited, together with a 40% decrease in IAA after 3 days of GR24 treatment. There were hundreds of differentially expressed genes (DEGs) with GR24 treatment, and in the DEGs list, the pathways of SL and cytokinin and the auxin signaling pathway were significantly enriched. Many DEGs were confirmed to be involved in the response of GR24-induced bud growth, especially for some transcription factors. In conclusion, our results not only provide new mechanistic insights into how SL regulates the axillary bud outgrowth of ratoon rice and provide a valuable resource for further functional characterization of axillary bud outgrowth in ratoon rice, but also lay the foundation for molecular breeding to improve rice ratooning ability.

## Figures and Tables

**Figure 1 plants-13-00899-f001:**
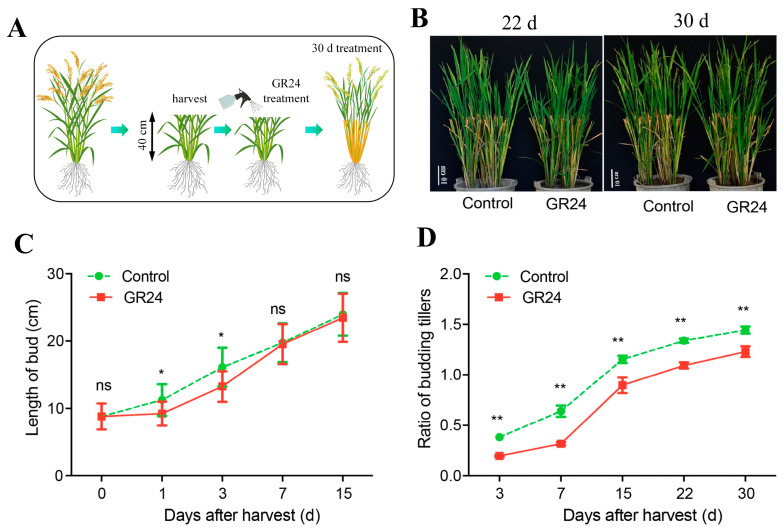
Performance of GR24 treatment induced axillary bud outgrowth of seasonal rice. (**A**) Flow chart representing the procedure of GR24 treatment at different stages of seasonal rice. (**B**) Plant performance of seasonal rice at different days after GR24 treatments. (**C**) Dynamics of GR24-induced length of bud in seasonal rice. (**D**) Dynamics of GR24-induced ration of budging tillers (ratoon rates). Data represent means ± SE ((**C**): *n* = 30; (**D**): *n* = 6). Symbols “*”, “**”, and “ns” indicated significant difference (*p* < 0.05), extremely significant difference (*p* < 0.01), and no significant difference, between GR24 treatment and control at the same treatment time, respectively.

**Figure 2 plants-13-00899-f002:**
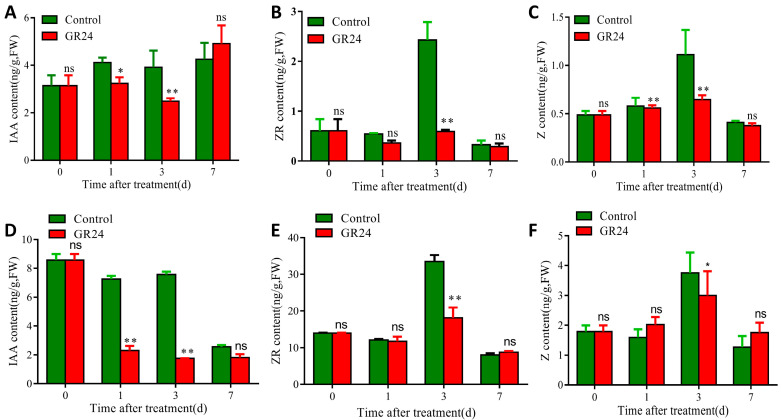
Effects of GR24 treatment on phytohormone contents in first seasonal rice. Panels (**A**–**C**) represent the IAA, ZR, and Z content in the second bud from the top of seasonal rice, respectively. Panels (**D**–**F**) represent the IAA, ZR, and Z content in the second section of the first seasonal rice, respectively. Symbols “*”, “**”, and “ns” indicate significant difference (*p* < 0.05), extremely significant difference (*p* < 0.01), and no significant difference between GR24 treatment and control at the same treatment time, respectively. *n* = 10.

**Figure 3 plants-13-00899-f003:**
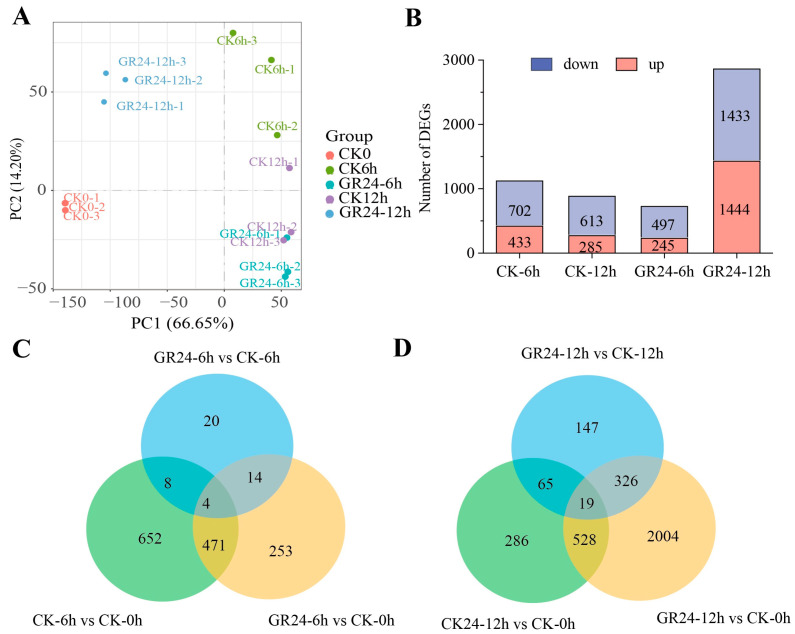
Differentially expressed gene analysis in 15 samples performed in this study. (**A**) Principal component analysis of the genes identified by transcriptomes. (**B**) Statistics of up- and downregulation of DEGs in different samples after 6 h and 12 h GR24 treatments relative to CK-0h. (**C**) Venn diagram representing the amount of up- and downregulation of DEGs for 6 h GR24 treatments. (**D**) Venn diagram representing the amount of up- and downregulation of DEGs for 12 h GR24 treatments.

**Figure 4 plants-13-00899-f004:**
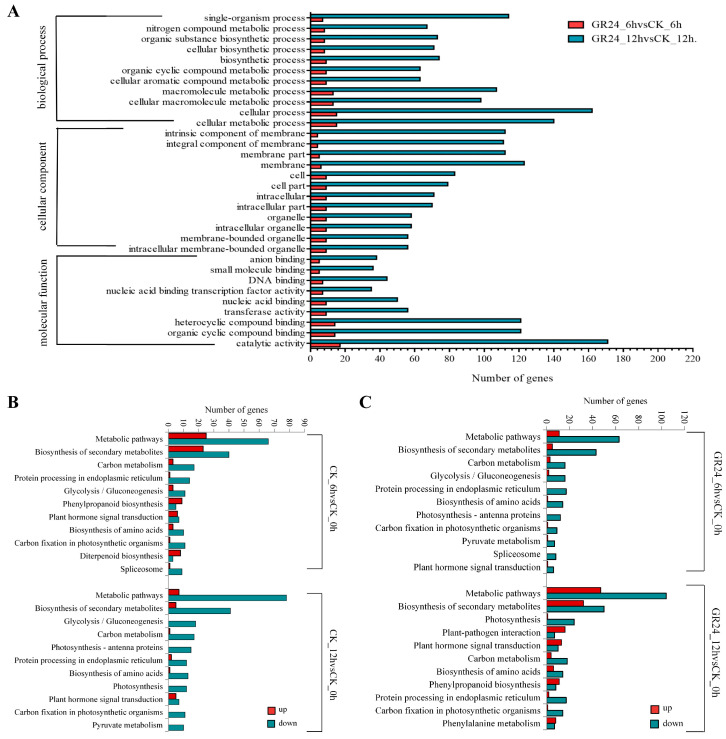
GO and KEGG pathway enrichment analysis on the list of DEGs in GR24 treatments for different durations relative to control in HHZ. (**A**) GO pathway enrichments of DEGs. (**B**) KEGG pathway enrichments of DEGs in comparisons of CK_6h vs. CK_0h and CK_12h vs. CK_0h. (**C**) KEGG pathway enrichments of DEGs in comparisons of GR24_6h vs. CK_0h and GR24_12h vs. CK_0h.

**Figure 5 plants-13-00899-f005:**
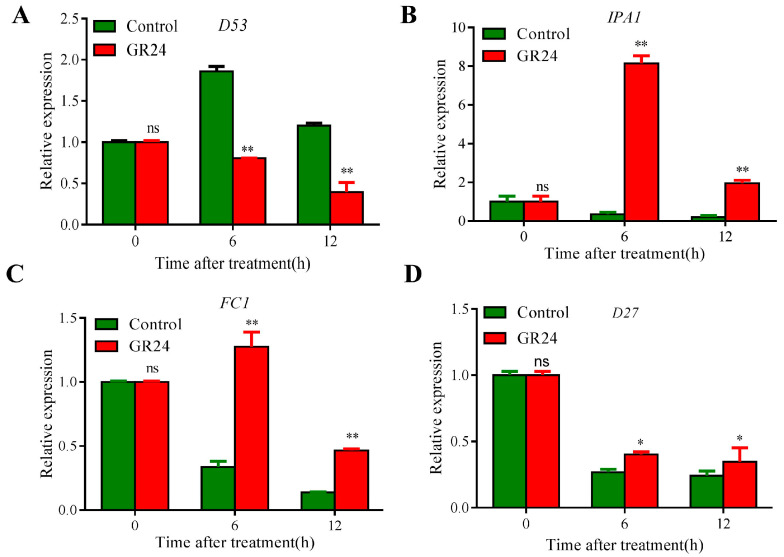
Gene expression involved in the strigolactone signaling pathway during the axillary bud outgrowth of seasonal rice with GR24 treatment. Panels (**A**–**D**) represent the relative gene expression levels of *D53*, *IPA1*, *FC1*, and *D27* at different times after GR24 treatments, respectively. Data represent means ± SE (*n* = 3). Expression value of 1 indicates a the gene expression level of 0 h as a control and converts the qPCR value of 0 h to 1. Symbols “*”, “**”, and “ns” indicate significant differences at *p* < 0.05 and *p* < 0.01 and no significant difference between GR24 treatment and control at the same treatment time, respectively.

**Figure 6 plants-13-00899-f006:**
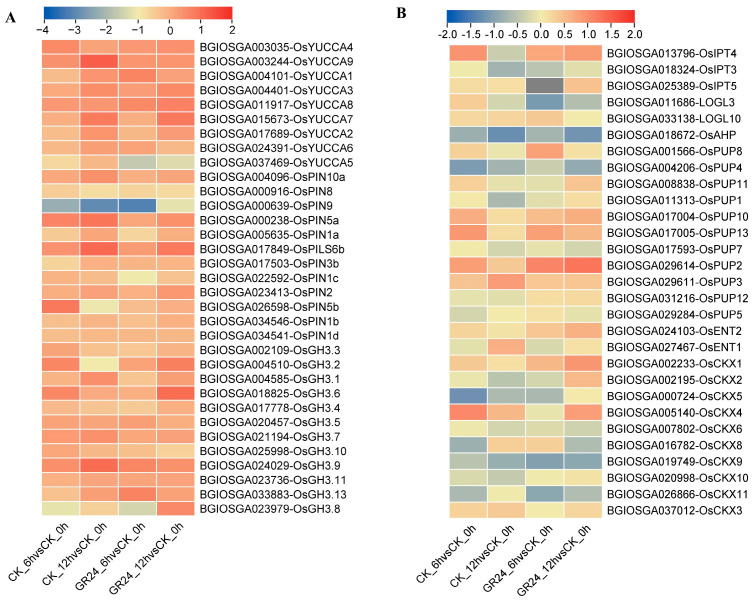
Comprehensive analysis of the gene expression of auxin and cytokinin homeostasis with GR24 treatment during the axillary bud outgrowth of seasonal rice. Panels (**A**,**B**) represent the heatmap representing the relative abundance of genes related to auxin and cytokinin biosynthesis pathways based on transcriptome, respectively. The relative expression levels are shown as log_2_Foldchange levels. Blue cells indicate downregulation and red cells indicate upregulation.

**Figure 7 plants-13-00899-f007:**
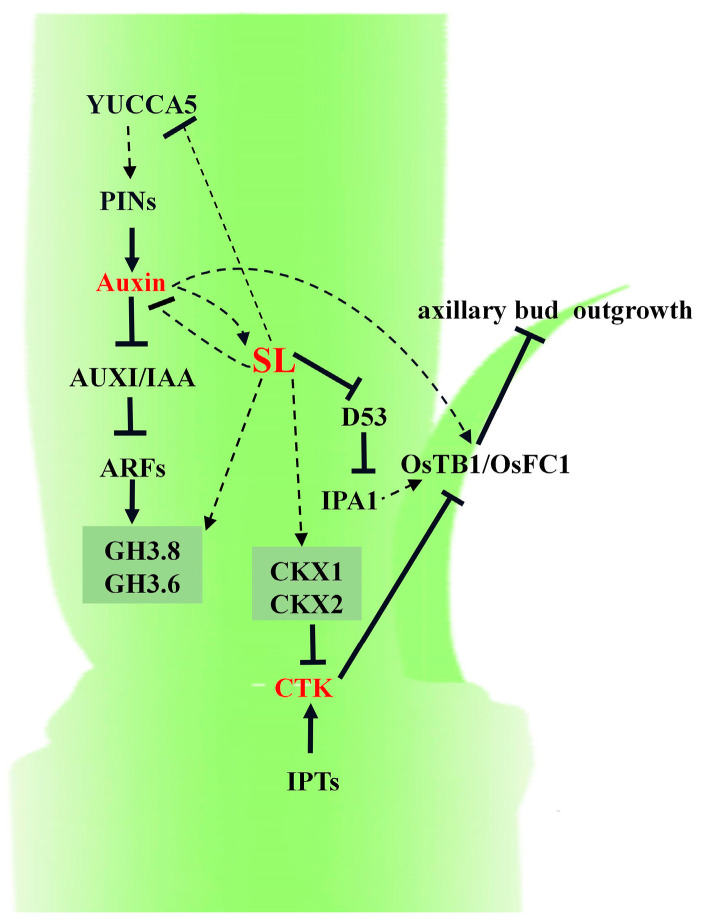
A simplified model for the regulation network on the axillary bud outgrowth of ratoon rice among strigolactone, cytokinin, and auxin. Arrowheads represent positive regulation; blunt arrows represent negative regulation; and dashed lines represent indirect regulation.

## Data Availability

Data are contained within the article and Appendix A.

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
