# Peer review of "Comparative Transcriptome Analysis Reveals Inhibitory Roles of Strigolactone in Axillary Bud Outgrowth in Ratoon Rice"

_plants, 2024, doi:10.3390/plants13060899_

Round 1
Reviewer 1 Report
Comments and Suggestions for Authors
In this article, a transcriptomic analysis is carried out in mouse rice plants in response to strigolactone looking at its effect on the growth of axillary bud. The results are interesting, although in my opinion little benefit has been gained from the work carried out. Perhaps the qPCR analysis could have been completed a little more.
Some comments regarding the manuscript:
- It is not clear to me the differences between strigolactone and the GR24 analogue, perhaps an explanatory line in the last introductory paragraph would clarify. This further questions the wisdom of using only strigolactone in the title.
- Since the materials and methods are at the end, it would help to indicate the abbreviations in the introduction to help the reader.
- Consider increasing the size of figures 4 and 5 using the left margin to help the reader of the manuscript.
- Figure 5, any explanation for the changes in gene expression when sprayed with water and determined in the control at 6 h? I don't know if the phrase on line 167 is the most appropriate: "the relative expression of .... were upregulated", if it does not change with time 0 when sprayed with GR24.
- Indicate in the figure legend or in materials and methods how the value of 1 is assigned in gene expression.
- There are several typing errors throughout the manuscript.
Author Response
Dear reviewers,
Thank you very much for your time to take care of our manuscript. Many thanks for your kindly comments and suggestions.
We have carefully revised the manuscript and addressed all the questions raised by the editor and reviewers recommendations. The important changes in text are also highlighted in the revised manuscript.
We would be grateful if our revised manuscript could be accepted for publication in Plants.
Sincerely yours,
Langtao Xiao
Professor
Lab director
Hunan Provincial Key Laboratory of Phytohormones and Growth Development Hunan Agricultural University
Changsha 410128, China
Email: ltxiao@hunau.edu.cn
Tel: 86-0731-84635261
- A. Reviewer 1’s comments and our responses
A1: In this article, a transcriptomic analysis is carried out in mouse rice plants in response to strigolactone looking at its effect on the growth of axillary bud. The results are interesting, although in my opinion little benefit has been gained from the work carried out. Perhaps the qPCR analysis could have been completed a little more.
Response to A1: Thanks for the comments and suggestion. Notably, we performed 3 replicates for each transcriptome sequencing sample and also rigorously controlled and analyzed the quality of sequencing data. When the sequencing results were obtained, we randomly selected some genes for qPCR validation, and most of the sequencing results were consistent with qPCR results. In addition, considering the maturity and reliability of current transcriptome sequencing technology, we did not exhibit the qPCR results in the article. Many thanks for your suggestion for qPCR analysis, we have attached the qPCR results (lines 154-158) which we have obtained as Figure S2 in supplementary materials. Please check!
A2: It is not clear to me the differences between strigolactone and the GR24 analogue, perhaps an explanatory line in the last introductory paragraph would clarify. This further questions the wisdom of using only strigolactone in the title.
Response to A2: Thanks for the question. GR24 is a strigolactone analogue, and widely been used by researchers in the analysis of SL network regulatory mechanisms [14-17]. Thanks for your comments and suggestion. In order to avoid misunderstandings, we have indicated in the abstract and introduction section (lines 12-13 and line 69).
A3: Since the materials and methods are at the end, it would help to indicate the abbreviations in the introduction to help the reader.
Response to A3: Thanks for the suggestion. We also agree with your suggestions for modifying the paper. We have indicated the abbreviations in the introduction section (lines 62- 68).
A4: Consider increasing the size of figures 4 and 5 using the left margin to help the reader of the manuscript.
Response to A4: Thank you again for your suggestions! We have made revision accordingly.We have adjusted Figures 4 and 5 to make them larger and clearer (line 178 and line 201).
A5: Figure 5, any explanation for the changes in gene expression when sprayed with water and determined in the control at 6 h? I don't know if the phrase on line 167 is the most appropriate: "the relative expression of .... were upregulated", if it does not change with time 0 when sprayed with GR24..
Response to A5: Thank you again for your suggestions! According to your suggestion, we have corrected this sentence to “As shown in Figure 5A, the expression levels of D53, which is an important repressor of SL signalling, were dramatically inhibited by GR24 (Figure 5A). Meanwhile, we found that Ideal Plant Architecture 1 (IPA1), a key regulator of the plant architecture in rice, functions as a direct downstream component of D53 in regulating tiller number and SL-induced gene expression, the expression of which is significantly induced within 12 h after GR24 treatment (Figure 5B). Furthermore, we found that the expression level of FC1, which works downstream of strigolactones to inhibit the outgrowth of axillary buds in rice, was also significantly higher than that of the control after GR24 treatment 6 h and 12 h (Figure 5C). In addition to, the relative expression of D27 was also significantly induced after GR24 treatment (Figure 5D). ” (lines 190 -199 ).
A6: Indicate in the figure legend or in materials and methods how the value of 1 is assigned in gene expression.
Response to A6: Thank you again for your suggestions! We use the gene expression level of 0 h as a control and convert the qPCR value of 0 h to 1. Therefore, the gene expression level at 6 and 12 hours after treatment is a relative expression level. We followed reviewer's suggestion to add description for the value of 1 in the figure legend in our revised manuscript (lines 205- 206).
A7: There are several typing errors throughout the manuscript.
Response to A7: Thanks for the suggestions. We have carefully checked our manuscript and revised accordingly.
Reviewer 2 Report
Comments and Suggestions for Authors
I believe this to be an interesting paper with a well thought out premise and experimental design. However, I think some of the data could be better explored. The discussion section is really short and some interesting insights could be taken from the rna-seq data. The authors could show functional enrichment for Up or Downregulated genes instead of just the global DEGs in order to grant some more insight into the mode of action of strigolactones here.
Author Response
Dear reviewers,
Thank you very much for your time to take care of our manuscript. Many thanks for your kindly comments and suggestions.
We have carefully revised the manuscript and addressed all the questions raised by the editor and reviewers recommendations. The important changes in text are also highlighted in the revised manuscript.
We would be grateful if our revised manuscript could be accepted for publication in Plants.
Sincerely yours,
Langtao Xiao
Professor
Lab director
Hunan Provincial Key Laboratory of Phytohormones and Growth Development Hunan Agricultural University
Changsha 410128, China
Email: ltxiao@hunau.edu.cn
Tel: 86-0731-84635261
- B. Reviewer 2’s comments and our responses
B1: I believe this to be an interesting paper with a well thought out premise and experimental design. However, I think some of the data could be better explored. The discussion section is really short and some interesting insights could be taken from the rna-seq data. The authors could show functional enrichment for Up or Down regulated genes instead of just the global DEGs in order to grant some more insight into the mode of action of strigolactones here.
Response to B1: Thank you for your comments and affirmation of this paper. We also agree with your suggestions for modifying the paper. In order to provide the readers with fair description of the facts, we have added appropriate content in the discussion section (line 264-276; line 288-302; line 314-323; line 326-334). Meanwhile, we have modified Figure 4 to show functional enrichment for up or down regulated genes in the KEGG pathways, which is better to show the machnism of SL on inhibition of axillary bud outgrowth in ratoon rice (line 178).
Reviewer 3 Report
Comments and Suggestions for Authors
Comments and Suggestions for Authors
Dear Author
I have an honor to review the manuscript entitled “Comparative transcriptome analysis reveals inhibitory roles of strigolactone in axillary bud outgrowth in ratoon rice” a research article submitted to the MDPI Journal, Plants. Authors of this manuscript performed transcriptome analysis in axillary bud from ratoon rice using molecular, gene expression and bioinformatic based study. They have treated strigolactone (SL) analogue GR24, to the ratoon rice just after first harvesting. Further performed expression analysis of genes related to plants growth and development. Overall, the experiments are performed well and the results are convincing. Thus, the presented results take up an important topic consistent with the profile of the Journal.
-However, even, manuscript is well organized and well described of the conception; I have some suggestions, which might improve the manuscript to make important to the wider audience.
-English should be improved throughout the text.
-Few suggestions I have mentioned in the main text pdf file. Please check
Abstract: -Good organization with results order. Need elaboration of the GR24 on its’ first-time use
1. Introduction
-The aim of the study should be underlined precisely and simultaneously and highlight why SL analogue GR24 treatment study is important in ratoon rice. Rationale to be elucidated for the purpose of the study.
-at least few discussions needed for GR24 application even in other plants
-Write something about bioinformatics study and their application in your research, because, research mostly on bioinformatics.
2. Results
L86; why “would” used? Check tense
Fig. 1; 30D at 1A and 1B do not match for panicle status. Change in 1A
-In 1B, you should have 0D (just after harvest and treatment)
-L106; what is graining stage? Is it scientific terminology?
-Fig. 4. Is not legible
3. Discussion
-More relative discussion needed based on results obtained, especially for gene expression in ratoon rice and others.
4. Materials and Methods
-What is the specificity of the HHZ rice?
-L284; What is “each group” indicates?
-L284-285; 1st time base fertilization, 2nd time tillering fertilizer, indicating they are different, but same NPK ratio. So, rewrite with different style
-Need brief description of phytohormone determination
5. Conclusion
What is the concluding suggestion from this research of SL treatment?

Minor editing of English language required
Author Response
Dear reviewers,
Thank you very much for your time to take care of our manuscript. Many thanks for your kindly comments and suggestions.
We have carefully revised the manuscript and addressed all the questions raised by the editor and reviewers recommendations. The important changes in text are also highlighted in the revised manuscript.
We would be grateful if our revised manuscript could be accepted for publication in Plants.
Sincerely yours,
Langtao Xiao
Professor
Lab director
Hunan Provincial Key Laboratory of Phytohormones and Growth Development Hunan Agricultural University
Changsha 410128, China
Email: ltxiao@hunau.edu.cn
Tel: 86-0731-84635261
- C. Reviewer 3’s comments and our responses
C1: English should be improved throughout the text.
Response to C1: We have carefully checked our manuscript and revised. We carefully checked throughout the text and standardized some language expressions which were highlighted in the revised manuscript.
C2: Few suggestions I have mentioned in the main text pdf file. Please check
Response to C1: Thanks for the suggestions. We also agree with your suggestions for modifying the paper. We have made revision accordingly(line 95; line 100; line 223; line 243; line 360; line 402).
C3: Abstract: Good organization with results order. Need elaboration of the GR24 on its’ first-time use
Response to C3: Thank you again for your comments and affirmation of this paper. We have add description for GR24 in the abstract section (lines 12-13).
C4: Introduction:
C4-1: The aim of the study should be underlined precisely and simultaneously and highlight why SL analogue GR24 treatment study is important in ratoon rice. Rationale to be elucidated for the purpose of the study.
Response to C4-1: Thanks for the suggestions. We also agree with your suggestions for modifying the paper. We followed your suggestion to revise our introduction (lines 71-87).
C4-2: at least few discussions needed for GR24 application even in other plants
Response to C4-2: Thanks for the suggestions. We followed your suggestion and have added appropriate content: “Mutations in the genes of SL biosynthesis lead to dramatically increased tiller number in rice, and this phenotype can be suppressed by exogenous treatment with the commonly used SL analog GR24 [14-17]”in lines 71-73. and “ GR24 is widely used in the study of the mechanism of SL controlling shoot branching in various plants, such as rice [18], pea [7], tomato [19] and tobacco [20]. ”in lines 85-87.
C4-3: Write something about bioinformatics study and their application in your research, because, research mostly on bioinformatics.
Response to C4-3: Excellent question! We fully agree with reviewer's suggestion. We have added appropriate content in lines 75-87: “ High-throughput sequencing technology has played an important role in revealing the molecular mechanisms of various biological processes in organisms. RNA sequencing (RNA-Seq) technology has been widely applied to assist in determining differentially expressed genes (DEGs) involved in different biological processes in many species and may be a promising method to address the genes associated with axillary bud outgrowth in ratoon rice. However, no studies on axillary bud in ratoon rice using RNA-Seq technology have been documented so far and the molecular mechanism regulating axillary bud outgrowth in ratoon rice is poorly understood. In the present study, to better reveal whether SL are involved in regulating the axillary bud outgrowth in ratoon rice, we used RNA-seq to identify genes in ratoon rice associated with axillary bud outgrowth after the application of GR24, which is widely used in the study of the mechanism of SL controlling shoot branching in various plants, such as rice[18], pea[7], tomato [19] and tobacco [20]. . ”
C5. Results
C5-1: L86; why “would” used? Check tense
Response to C5-1: Thanks for the suggestions. We have deleted the word of the would in this sentence (line 100).
C5-2:Fig. 1; 30D at 1A and 1B do not match for panicle status. Change in 1A
Response to C5-2: Thanks for the suggestions. We have changed Figures 1A (line 108 ).
C5-3:-In Fig 1B, you should have 0D (just after harvest and treatment)
Response to C5-3: Thanks for the suggestions. We are terribly sorry that we forgot to take the 0 d photo due to being busy with processing and collecting samples at that time to ensure the validity of the sample. However, in order to ensure the rigor of the experiment, we strictly followed method 4.2 for sampling and spraying GR24 treatment after the first season harvesting. This method ensures all materials with consistent status, including the pile height and spraying process. And we can also see that the growth and pile height of the first season rice piles in the 22 d and 30 d treatments and the control are consistent from Figure 1B. All in all, we deeply apologize for our mistake. We accept criticisms and hope to avoid similar mistakes by emphasize.
C5-4:L106; what is graining stage? Is it scientific terminology?
Response to C5-4: Thanks for the suggestions. I am terribly sorry for this expression error of graining stage. It should be milk-ripe stage. We have corrected this phrase in line 122.
C5-5: Fig. 4. Is not legible
Response to C5-5: Thank you again for your suggestions! We have replaced figure 4 with higher resolution (line 178).
C6. Discussion
C6-1: More relative discussion needed based on results obtained, especially for gene expression in ratoon rice and others.
Response to C6-1: Thank you for your suggestions. We also agree with your suggestions for modifying the paper. In order to provide the readers with fair description of the facts, we have added appropriate content in the discussion section (line 264-276; line 288-302; line 314-323; line 326-334).
C7: Materials and Methods
C7-1: What is the specificity of the HHZ rice?
Response to C7-1: Thank you for your suggestions. HHZ is a traditional type of rice.We chose HHZ because it is a high-quality conventional indica rice, which is characterized with strong ratooning ability, high yield in production, fertilizer tolerance, lodging resistance and it is widely cultivated as a ratoon rice in southern China. We have added the explanation to the text (lines 346-348 ).
C7-2: L284-285; 1st time base fertilization, 2nd time tillering fertilizer, indicating they are different, but same NPK ratio. So, rewrite with different style
Response to C7-2: Thanks for the suggestions. We have revised the manuscript in lines 350 -353 as “ 6 g of balanced base fertilizer(N:P:K = 15:10:10) were applied for each pot at 7 d before transplanting, 3 g of tillering fertilizer (N:P:K = 15:10:10) were also applied to each pot at tillering stage.”
C7-3:Need brief description of phytohormone determination
Response to C7-3: Thanks for the comments. We have added the phytohormone determination in our revised manuscript (lines 379-401).
C8. Conclusion What is the concluding suggestion from this research of SL treatment?
Response to C8-1: Thank you again for your suggestions! We have revised the conclusion section to highlight some key findings and suggestions of this article (lines 463-467).
Round 2
Reviewer 2 Report
Comments and Suggestions for Authors
My comments have been addressed.